# The Impact of Psychological Well-Being on Learning Strategies: Analyzing Perceived Stress, Self-Esteem, and Study Approaches in Nursing and Obstetrics Students

**DOI:** 10.3390/nursrep15030109

**Published:** 2025-03-19

**Authors:** Antonietta Pacifico, Luisa Gorrese, Carlo Sorrentino, Michele Viciconte, Vincenzo Andretta, Paola Iovino, Giulia Savarese, Carolina Amato, Luna Carpinelli

**Affiliations:** 1Department of Medicine, Surgery and Dentistry, Scuola Medica Salernitana, University of Salerno, 84081 Baronissi, Italy; apacifico@unisa.it (A.P.); lgorrese@unisa.it (L.G.); casorrentino@unisa.it (C.S.); mviciconte@unisa.it (M.V.); vandretta@unisa.it (V.A.); piovino@unisa.it (P.I.); lcarpinelli@unisa.it (L.C.); 2Master’s Degree Course in Organizational Psychology, Marketing, and Human Resources, Università Cattolica del Sacro Cuore, 20123 Milan, Italy; carolina.amato02@icatt.it

**Keywords:** academic motivation, nursing students, perceived stress, psychological well-being, self-esteem

## Abstract

**Background:** The psychological well-being of university students significantly impacts their academic performance and future professional preparation. The aim of this study is to analyze the relationships between perceived stress, self-esteem, and learning strategies in university students enrolled in Nursing and Obstretics degree programs, in order to understand the impact of psychological well-being on their study abilities. **Method:** This study is observational cross-sectional, using non-probabilistic convenience sampling. The study involved students enrolled in Nursing and Obstetrics courses at the University of Salerno. A standardized self-reported questionnaire will be used. **Results:** 331 students (82.75%) participated voluntarily. Correlations show significant relationships consistent with psychological literature. Higher self-esteem correlates with lower perceived stress (r = −0.325, *p* < 0.01), better information elaboration (r = 0.156, *p* < 0.01), and higher metacognitive awareness (r = 0.123, *p* < 0.05), but negatively with organizational strategies (r = −0.150, *p* < 0.01) and self-evaluation frequency (r = −0.153, *p* < 0.01). Perceived stress correlates positively with organizational strategies (r = 0.180, *p* < 0.01) and self-evaluation frequency (r = 0.178, *p* < 0.01), suggesting stress may drive compensatory strategies. Multiple regression analyses showed that self-esteem was a significant positive predictor of information elaboration strategies (β = 0.49, *p* = 0.05). Both self-esteem (β = −0.52, *p* = 0.01) and perceived stress (β = −0.74, *p* = 0.01) were significant negative predictors of structured learning strategies, suggesting that higher levels of stress and self-esteem are associated with a reduced use of planned organizational approaches. The models explained a substantial proportion of variance, with adjusted R^2^ values of 0.52 for elaboration and 0.63 for strategy components. **Conclusions:** These results emphasize the need for interventions to enhance learning strategies and stress management among students.

## 1. Introduction

University students’ psychological well-being directly impacts their academic performance and the quality of their professional training [1]. Students’ mental health not only affects their ability to face academic challenges but can also determine their future success as healthcare professionals [2]. Nursing students face a significantly demanding study load, including theoretical learning and complex practical experiences such as clinical internships and group work in healthcare settings characterized by high stress levels. These challenges are often accompanied by strong emotional pressure arising from the difficulties in dealing with critical situations, such as providing care to patients in emergency conditions or treating severe illnesses, which can significantly affect their motivation and stress management [3,4].

Perceived stress, time management, and study motivation are key factors that influence academic performance. However, when high stress levels hinder these factors, students become more susceptible to burnout, a phenomenon characterized by emotional exhaustion due to their inability to cope with the high demands of academic work. Burnout not only reduces motivation and study effectiveness but can also compromise the quality of interactions with teachers and classmates. The relationship between motivation and burnout can create a vicious cycle, where stress and low motivation further hinder time management, worsening academic performance [5,6].

Research indicates that low self-esteem can distort students’ perception of their abilities, perpetuating stress and reducing motivation [7]. Conversely, good self-esteem and a positive self-perception enhance self-efficacy, facilitating stress management and improving study motivation [8,9].

Given the high academic workload and emotional demands in nursing education, understanding these dynamics is essential. Excessive stress can hinder effective study strategies, leading to poor performance and a higher risk of burnout, while higher self-esteem supports better learning strategies, enabling students to tackle academic challenges more effectively.

Despite the growing body of research on the psychological well-being of university students, there are still significant gaps in understanding the specific dynamics at play within nursing education. Much of the existing literature focuses on individual factors such as stress, self-esteem, and motivation, but there is a lack of comprehensive studies examining how these factors interact within the unique context of nursing programs. Furthermore, while studies on burnout in students are common, few explore the direct link between self-esteem and time management strategies in the context of nursing students’ academic challenges. Additionally, research examining the effectiveness of targeted interventions aimed at improving self-esteem and stress management in nursing students remains limited.

This study aims to fill these gaps by exploring the interplay between learning strategies, perceived stress, and self-esteem specifically within nursing students. While the correlation between these factors is well established, identifying their unique impact in this context can provide valuable insights for developing targeted strategies to enhance students’ academic performance and well-being in the future.

## 2. Materials and Methods

### 2.1. Research Design

This study employed cross-sectional descriptive correlational research design using a non-probabilistic convenience sampling method. The aim was to analyze the relationships between learning strategies, perceived stress, and self-esteem among university students.

### 2.2. Participants

A total of 331 students (82.75%) enrolled in the Bachelor’s degree program in Nursing and Obstetrics Sciences at the University of Salerno participated voluntarily in the study. The sample consisted of 70.7% female students, with a mean age of 23 years (range: 18–40 years). The majority of participants (65%) were aged between 20 and 25 years.

### 2.3. Sampling Design

A non-probabilistic convenience sampling method was employed. Eligibility criteria included (a) being actively enrolled in the Nursing and Obstetrics program, (b) being at least 18 years old, and (c) providing informed consent for participation.

The required sample size was determined based on an a priori power analysis following Cohen’s guidelines [10]. With an expected medium effect size of 0.30 (f^2^ = 0.15), a power of 80%, and a significance level of 0.05, a minimum of 68 participants was required to detect statistically significant effects in multiple regression analyses with two predictors.

The final sample consisted of 331 participants, ensuring adequate statistical power.

### 2.4. Instruments

Standardized and validated instruments were used for data collection:

1. The Study Approach Questionnaire (QAS—AMOS battery) [11] is a standardized questionnaire designed to assess students’ learning strategies. It includes five subscales: Organization, Elaboration, Self-Evaluation, Strategy, and Metacognition.

Higher scores on each subscale indicate more frequent and effective use of study skills and strategies. For example, higher scores on the Elaboration scale reflect deeper processing and integration of learning materials, while higher Metacognition scores suggest better self-monitoring and regulation during study activities.

The psychometric properties of the QAS are well established. Internal consistency (Cronbach’s α) ranges from 0.71 to 0.89, depending on the subscale. Construct validity has been confirmed through factor analysis, supporting its multidimensional structure. The questionnaire also demonstrates good convergent and discriminant validity with related constructions, such as academic motivation and study habits.

2. Perceived Stress was measured using the 10-item Perceived Stress Scale (PSS-10) [12], which assesses stress levels over the past 30 days. Responses are recorded on a 5-point Likert scale, with higher scores indicating greater perceived stress. The scale has demonstrated good internal consistency (Cronbach’s α = 0.78–0.91).

3. Self-Esteem was measured using the 10-item Rosenberg Self-Esteem Scale (RSES) [13], which assesses global self-esteem on a 4-point Likert scale. Higher scores indicate higher self-esteem. This scale has demonstrated strong reliability (Cronbach’s α = 0.77–0.88).

### 2.5. Procedure

The study was conducted between October and December 2024. Students were invited via email to participate, and recruitment was supervised by the President of the Academic Council with the support of course assistants. Data collection was conducted using an anonymous Google Forms questionnaire, which required approximately 40 min to complete. To address power imbalances, participation was voluntary, and anonymity was ensured [14].

### 2.6. Data Analysis

Data were analyzed using IBM SPSS v.23 software. Descriptive statistics (means, standard deviations, ranges) were calculated for each construct, and normality tests were performed. To explore relationships among study variables, the following analyses were conducted: (a) bivariate Pearson correlations to examine associations between learning strategies, perceived stress, and self-esteem; and (b) multiple regression analysis to determine the predictive power of self-esteem and perceived stress on learning strategies and motivation.

### 2.7. Ethical Consideration

This study was conducted according to the Declaration of Helsinki (1964) and the recommendations of the “Associazione Italiana di Psicologia” (AIP). This study was approved by the local Ethics Committee “Centro di Counseling Psicologico” (number 01/2021).

## 3. Results

### 3.1. Learning Strategies

The analysis of learning strategies among students highlights differences in various aspects of study approaches (see Table 1). Organizational strategies were found to be moderate, with an average score of 2.77 (SD = 0.408), indicating a general consistency among participants in their ability to structure study activities. The ability to process information, as measured by elaboration strategies, showed a higher mean score of 3.46 (SD = 0.586), reflecting good competence in integrating and understanding new knowledge, although variability in responses suggests differences in approach among students.

Self-evaluation, which refers to the tendency to assess one’s learning progress, yielded a moderate average score of 2.73 (SD = 0.373), with low variability in responses. The adoption of general study strategies was relatively low (M = 2.26, SD = 0.360), suggesting a lack of structured planning and resource utilization. Finally, metacognitive awareness, which reflects the ability to monitor and regulate one’s learning process, showed a higher mean score of 3.47 (SD = 0.526), indicating that students generally exhibit a good level of reflection and self-regulation, though with some individual differences.

### 3.2. Perceived Stress

Perceived stress among students was assessed using the Perceived Stress Scale (PSS-10), which yielded an average score of 20 (SD = 5.465). These results indicate a moderate level of stress, with variability in responses suggesting that some individuals experience higher levels of psychological distress, while others report lower stress levels.

### 3.3. Self-Esteem

The self-esteem levels of students were evaluated using the Rosenberg Self-Esteem Scale, which resulted in a mean score of 19.30 (SD = 4.821). This indicates a moderate level of self-esteem, with scores distributed within a relatively stable range, suggesting that while some students may have lower self-confidence, extreme deviations from the mean are limited.

### 3.4. Correlations Between Variables

The correlation analysis (see Table 2) reveals significant relationships between self-esteem, perceived stress, and learning strategies. Higher self-esteem is associated with lower perceived stress (r = −0.325, *p* < 0.01), stronger information processing strategies (r = 0.156, *p* < 0.01), and greater metacognitive awareness (r = 0.123, *p* < 0.05). However, self-esteem is negatively correlated with the use of structured organizational strategies (r = −0.150, *p* < 0.01) and self-evaluation frequency (r = −0.153, *p* < 0.01), suggesting that students with greater self-confidence tend to rely less on structured learning approaches.

Perceived stress is positively associated with the adoption of organizational strategies (r = 0.180, *p* < 0.01) and self-evaluation (r = 0.178, *p* < 0.01), indicating that students experiencing stress may develop compensatory strategies to manage academic demands.

Furthermore, the relationships among learning strategies underscore their interconnectedness: organizational strategies strongly correlate with self-evaluation (r = 0.640, *p* < 0.01), while information elaboration is positively related to metacognitive awareness (r = 0.616, *p* < 0.01). These findings highlight the importance of an integrated learning approach, where different strategies reinforce each other to support academic success.

### 3.5. Regression Analysis

To further explore the relationships between perceived stress, self-esteem, and learning strategies, a multiple regression analysis was conducted for each dimension of the Study Approach Questionnaire (QAS). The independent variables included self-esteem (RSES) and perceived stress (PSS-10).

The results (Table 3) show that self-esteem was a significant predictor of the *QAS Elaboration* dimension (β = 0.49, *p* = 0.05), suggesting that students with higher self-esteem tend to use more sophisticated cognitive elaboration strategies. Perceived stress was not a significant predictor for this component (β = 0.32, *p* = 0.06).

For the *QAS Strategy* dimension, both self-esteem (β = −0.52, *p* = 0.01) and perceived stress (β = −0.74, *p* = 0.01) were significant predictors. This suggests that higher levels of stress and self-esteem are associated with a reduced use of structured organizational strategies.

In the other QAS dimensions (*Organization*, *Self-Evaluation*, and *Metacognition*), no predictors reached statistical significance. For example, perceived stress showed a positive but non-significant association with *Organization* (β = 0.28, *p* = 0.07) and *Self-Evaluation* (β = 0.26, *p* = 0.09). Self-esteem showed a positive but non-significant association with *Metacognition* (β = 0.24, *p* = 0.07).

The adjusted R^2^ values indicate that the models explained a substantial percentage of variance, particularly for *QAS Elaboration* (adjusted R^2^ = 0.52) and *QAS Strategy* (adjusted R^2^ = 0.63).

## 4. Discussion

The findings of this study highlight the crucial role of stress and self-esteem in shaping learning behaviors among nursing students, reinforcing the impact of psychological well-being on academic performance [15]. By examining the relationships between these variables, the study demonstrates that higher levels of stress tend to correlate with increased reliance on compensatory learning strategies, while greater self-esteem is associated with more effective cognitive engagement. These insights emphasize the importance of psychological factors in academic success within the nursing education context.

A key implication of these results is the necessity to integrate stress management and self-confidence enhancement into nursing education programs. Developing interventions aimed at reducing stress levels and fostering self-esteem could positively influence students’ self-regulated learning strategies, ultimately enhancing their ability to absorb, process, and apply knowledge effectively in both academic and professional settings.

In analyzing the components of self-regulated learning, it emerges that students exhibit moderate levels of organization in their study routines, indicating a structured but not highly developed approach to academic tasks. Organizational strategies play a fundamental role in managing coursework efficiently, as previous research suggests that well-structured study methods are positively linked to academic performance [16,17]. However, the limited variability in these results indicates that most students follow a similar organizational pattern, potentially limiting the effectiveness of interventions targeting this strategy.

Regarding elaboration strategies, the results suggest that students display a good ability to process and integrate new information, an essential strategy for deep learning [18,19]. However, the observed variability among participants suggests that some students may lack the cognitive strategies necessary for advanced information processing. This reinforces the need for targeted educational approaches that promote deeper engagement with learning materials, enhancing critical thinking and problem-solving strategies.

Finally, the study highlights the importance of metacognitive awareness in learning, as students who actively reflect on their cognitive processes tend to exhibit stronger self-regulation. Metacognition is a key factor in academic success, allowing students to monitor their own learning progress and adjust their strategies accordingly [20]. However, the variability observed in this dimension suggests that not all students engage in metacognitive reflection at the same level, which may impact their ability to optimize learning strategies effectively [21,22].

These findings support the need for a comprehensive educational framework that goes beyond traditional curriculum design and incorporates psychological well-being strategies to enhance students’ academic experiences. Future research should further explore interventions aimed at strengthening self-regulation and emotional resilience, ensuring that nursing students are better equipped to face both academic challenges and the demands of their future professional roles.

## 5. Limitations

The findings of this study suggest that, while students demonstrate essential self-regulation strategies, the variability in scores for elaboration and metacognition indicates that there are opportunities for improvement in these areas. Educational interventions targeting these dimensions could enhance students’ learning strategies and overall academic performance [23,24,25].

The study’s cross-sectional design, while useful for an initial analysis, limits the ability to establish direct effects on academic performance. Additionally, while self-esteem (r = 0.325) is negatively correlated with perceived stress and positively correlated with metacognition and elaboration, it is important to note that these relationships do not necessarily imply that changes in self-esteem lead to improvements in academic outcomes.

Finally, the lack of direct assessment of academic performance represents another limitation, suggesting the need to incorporate additional variables, such as the effectiveness of study strategies or the learning environment, to provide a more comprehensive understanding of the factors influencing academic success.

## 6. Conclusions

Previous studies have shown that self-esteem can influence academic outcomes [1,2,23]. While this study is preliminary and serves as an initial analysis, the results suggest that self-esteem and stress are significant factors that may influence how students engage with their learning processes and manage academic challenges [26]. This study provides an important first step in understanding these factors and paves the way for future, more in-depth research.

Future research could build on these findings by incorporating direct measures of academic performance and exploring how changes in self-regulation, stress, and self-esteem over time influence students’ learning outcomes. Longitudinal studies could provide more robust evidence of causal relationships and offer insights into the long-term impact of these variables on academic success.

## Figures and Tables

**Table 1 nursrep-15-00109-t001:** Descriptive Statistics of study variables.

Variables	M	SD
Study skill: organization	2.77	0.408
Study skill: elaboration	3.45	0.586
Study skill: self-evaluation	2.73	0.373
Study skill: strategy	2.26	0.360
Study skill: metacognition	3.47	0.526
Perceived stress	20	5.465
Self-esteem	19.30	4.821

**Table 2 nursrep-15-00109-t002:** Significant correlations among variables in the sample (N = 331).

Variable	Self-Esteem	Perceived Stress	Study Skill: Organization	Study Skill: Organization	Study Skill: Organization	Study Skill: Organization	Study Skill: Organization
Self-esteem							
Perceived stress	−0.325 *						
Study skill: organization	−0.150 **	0.180 **					
Study skill: elaboration	0.156 **	0.090	0.298 **				
Study skill: self-evaluation	−0.153 **	0.178 **	0.640 **	0.191 **			
Study skill: strategy	−0.029	0.018	0.487 **	0.164 **	0.464 *		
Study skill: metacognition	0.123 *	0.097	0.406 **	0.616 **	0.424 **	0.386 **	

Correlation is significant at the 0.01 level (two-tailed) **. Correlation is significant at the 0.05 level (two-tailed) *.

**Table 3 nursrep-15-00109-t003:** Multiple regression analysis predicting learning strategies.

Dependent Variable (QAS Dimension)	β (Self-Esteem)	*p* (Self-Esteem)	β (Perceived Stress)	*p* (Perceived Stress)	Adjusted R^2^
Organization	−0.22	0.08	0.28	0.07	0.32
Elaboration	** *0.49* **	** *0.05* **	0.32	0.06	0.52
Self-Evaluation	−0.18	0.12	0.26	0.09	0.29
Strategy	**−0.52**	**0.01**	**−0.74**	**0.01**	0.63
Metacognition	0.24	0.07	0.14	0.09	0.35

Note: β = Standardized regression coefficient. *p*-values indicate statistical significance. **Bold** values indicate significance at *p* < 0.01; italicized values indicate significance at *p* < 0.05. Adjusted R^2^ = proportion of variance explained by the model, adjusted for the number of predictors.

## Data Availability

The data that supports the findings of this study are available from the corresponding author (G.S.) upon reasonable request.

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
