# Peer review of "The Impact of Psychological Well-Being on Learning Strategies: Analyzing Perceived Stress, Self-Esteem, and Study Approaches in Nursing and Obstetrics Students"

_nursrep, 2025, doi:10.3390/nursrep15030109_

Round 1
Reviewer 1 Report
Comments and Suggestions for Authors
Dear Editor
Thank you for the opportunity to review the manuscript entitled “Stress, self-esteem, and study strategies: correlation analysis of variables influencing academic performance in nursing stu-dents” submitted for publication in the Nursing reports journal. The purpose of this manuscript is to analyse the interactions between study skills, motivation, perceived stress, and self-esteem in a sample of students. The authors administer a series of instruments to students on the Bachelor of Science in Nursing and Master of Science in Nursing and Midwifery course at the University of Salerno, analysing the relationships between the concepts measured. The main results are that high self-esteem is associated with lower perceived stress (r= -0.325, p< 0.01), higher information processing (r= 0.156, p< 0.01) and higher metacognitive awareness (r= 0.123, p< 0.05). Although, it is negatively correlated with the use of organisational strategies (r= -0.150, p< 0.01) and with the frequency of self-assessment (r= -0.153, p< 0.01). Whereas perceived stress (PSS-10) correlated positively with the use of organisational strategies (r= 0.180, p< 0.01) and the frequency of self-assessment (r= 0.178, p< 0.01). Finally, the authors conclude that low self-esteem is associated with difficulties in academic performance, whereas good stress management and the adoption of metacognitive study strategies contribute positively to well-being and learning effectiveness.
The topic is very interesting and important. It would be of interest to readers of Nursing reports, especially in view of the global shortage of nurses and the challenges related to their education. However, this manuscript requires major refinement and reorganisation for publication.
Major Considerations
1. Introduction
1.1 At the end of the introduction, the authors state that “the study intends to develop recommendations for targeted interventions that can support students in stress management, burnout prevention and improvisation of their motivation, in order to foster their commitment to their studies and, consequently, their professional training”. However, these recommendations are not developed. If the authors agree, I suggest deleting this paragraph.
2. Materials and method
2.1. The design of the applied study is not stated. Therefore, it becomes difficult to assess whether it is appropriate to address the research objective/purpose.
2.2. Could more context be provided in relation to the study settings and the recruitment of participants? In what period was it conducted? How many participants were invited compared to those who agreed to participate?
2.3. Recruitment procedures should be described – how and who recruited the participants for this research?
2.4. The participants are students so there is some degree of power imbalance between the participants and the researchers (especially if the researchers are educators). Could the authors comment on this aspect of the research? Specifically, how was coercion mitigated? How was power imbalance managed? This resource may be helpful:
Ferguson, L. M., Myrick, F., & Yonge, O. (2006). Ethically involving students in faculty research. Nurse education today, 26(8), 705–711. https://doi.org/10.1016/j.nedt.2006.07.021
2.5. The authors do a good job of reporting the instruments used to measure the concepts. However, is there any psychometric data to report on the instruments administered?
2.6. Participants. Besides gender and age, is it possible to have more information on the sample enrolled? Did you collect any other variables?
2.7. The authors state that ‘Students with documented clinical diagnoses of severe psychiatric disorders that could compromise the reliability of the data collected were not included in the study”. This statement leaves me very puzzled. Could the authors comment on this aspect of the research? Specifically, how was it assessed whether a student had a diagnosis of a psychiatric disorder? Also, is it ensured that no bias was introduced for this variable? Are you sure you can guarantee this?
2.8. I am not a statistician; however, the statistical analysis descriptions seem appropriate.
3. Results
3.1. The results are difficult to follow. I suggest including a figure for a clearer understanding of the values of the instruments used and their different components
3.2. The authors do a good job in describing the correlations between the variables.
4. Discussion
4.1. The discussion should be reviewed and expanded. The authors report their results; however, they do not discuss them in the context of the existing literature. The authors propose some explanations for their results. It is useful here to discuss the sub-elements of the concepts (e.g. ‘Organisation’, ‘Processing’ and ‘Metacognition’) and this could be strengthened if these sub-elements were introduced first in the paper and then expanded upon.
4.2. In the first part of the discussion, the authors state that the variables studied ‘significantly influence academic performance’. I believe this is incorrect; the results obtained do not support this statement.
5. Conclusion
5.1. The conclusions should be revised not entirely in line with the results. Where does it follow that low self-esteem is associated with difficulties in academic performance?
General Minor Considerations
- -the title does not correspond to the objective in the abstract: the concepts/variables should be the same. The academic performance of nursing students is not measured in the study
- “standardized and scientifically validated instruments”. I think it could be expressed in clearer terms.
- Revise the bibliographical references, they do not coincide with the citations within the text.
Author Response
The Authors thank you for the excellent suggestions, which improved the paper. In the paper, the corrections are highlighted in yellow.
- Introduction
1.1 At the end of the introduction, the authors state that “the study intends to develop recommendations for targeted interventions that can support students in stress management, burnout prevention and improvisation of their motivation, in order to foster their commitment to their studies and, consequently, their professional training”. However, these recommendations are not developed. If the authors agree, I suggest deleting this paragraph.
Authors R: The Authors thank the Reviewer for the insights and suggestions to improve our work. All changes are highlighted in yellow.
Regarding point 1.1, we have removed the paragraph and revised the introduction, highlighting the suggested aspects related to the objectives and aims of the study.
2. Materials and method
2.1. The design of the applied study is not stated. Therefore, it becomes difficult to assess whether it is appropriate to address the research objective/purpose.
2.2. Could more context be provided in relation to the study settings and the recruitment of participants? In what period was it conducted? How many participants were invited compared to those who agreed to participate?
2.3. Recruitment procedures should be described – how and who recruited the participants for this research?
2.4. The participants are students so there is some degree of power imbalance between the participants and the researchers (especially if the researchers are educators). Could the authors comment on this aspect of the research? Specifically, how was coercion mitigated? How was power imbalance managed? This resource may be helpful:
Ferguson, L. M., Myrick, F., & Yonge, O. (2006). Ethically involving students in faculty research. Nurse education today, 26(8), 705–711. https://doi.org/10.1016/j.nedt.2006.07.021
Authors R: For points 2.1, 2.2, 2.3, and 2.4, we have completely revised Section 2, incorporating the missing information and removing certain elements that could potentially confuse the reader (see paragraph 2.1).
2.5. The authors do a good job of reporting the instruments used to measure the concepts. However, is there any psychometric data to report on the instruments administered?
Authors R: In paragraph 2.2, we have specified the standardized nature of the selected tests, and their psychometric properties are already indicated in the references. Unfortunately, we cannot elaborate further, as our article is presented as a brief report with a limited character count. We believe it is more useful to address these aspects in the Results and Conclusions sections.
2.6. Participants. Besides gender and age, is it possible to have more information on the sample enrolled? Did you collect any other variables?
Authors R: We have not collected any further variables at the moment and have included this point as a limitation of our study.
2.7. The authors state that ‘Students with documented clinical diagnoses of severe psychiatric disorders that could compromise the reliability of the data collected were not included in the study”. This statement leaves me very puzzled. Could the authors comment on this aspect of the research? Specifically, how was it assessed whether a student had a diagnosis of a psychiatric disorder? Also, is it ensured that no bias was introduced for this variable? Are you sure you can guarantee this?
Authors R: Given the nature of our study and the survey variables, we thought about eliminating this specification which objectively only confuses a piece of data that is not relevant. Thank you!
2.8. I am not a statistician; however, the statistical analysis descriptions seem appropriate.
Authors R: Thank you!!
3. Results
3.1. The results are difficult to follow. I suggest including a figure for a clearer understanding of the values of the instruments used and their different components
Authors R: Table 1 of descriptive analysis has been included.
3.2. The authors do a good job in describing the correlations between the variables.
Authors R: Thank you!!
4. Discussion
4.1. The discussion should be reviewed and expanded. The authors report their results; however, they do not discuss them in the context of the existing literature. The authors propose some explanations for their results. It is useful here to discuss the sub-elements of the concepts (e.g. ‘Organisation’, ‘Processing’ and ‘Metacognition’) and this could be strengthened if these sub-elements were introduced first in the paper and then expanded upon.
4.2. In the first part of the discussion, the authors state that the variables studied ‘significantly influence academic performance’. I believe this is incorrect; the results obtained do not support this statement.
5. Conclusion
5.1. The conclusions should be revised not entirely in line with the results. Where does it follow that low self-esteem is associated with difficulties in academic performance?
Authors R: We revised the paragraphs Discussion, Limitations and Conclusions based on both your valuable suggestions and those of Reviewer 2.
General Minor Considerations
- -the title does not correspond to the objective in the abstract: the concepts/variables should be the same. The academic performance of nursing students is not measured in the study
Authors R: We have reworded the abstract to better align it with the title and objectives of the study.
- “standardized and scientifically validated instruments”. I think it could be expressed in clearer terms.
Auhtors R: We have revisited this sentence.
- Revise the bibliographical references, they do not coincide with the citations within the text.
Authors R: We performed a thorough review of the bibliographical references and in-text citations, ensuring that each citation is correctly stated in the bibliography, and vice versa, so as to maintain consistency between the body of the text and the sources used.
Reviewer 2 Report
Comments and Suggestions for Authors
This article is very well written and provides confirmation of the impact of stress and self efficacy on study behaviors in nursing students. The tools selected measure these constructs and the methods are appropriate to the research question and aims. The conclusion does not over reach but appropriately suggests that interventions be designed to support students based on these findings. For more comments, please see the attachment

Author Response
The Authors thank you for the excellent suggestions, which improved the paper. In the paper, the corrections are highlighted in blue.
main question addressed by the research:
Provides confirmation of the impact of stress and self efficacy on study behaviors in nursing students (the purpose of the study)
I do not find the article to be original except for its study of these particular students. It does not address a gap in the literature. The relationship between these variables is well understood.
subject area compared with other published material:
Brings to light the relationship between these variables in a nursing context.
The conclusion does not overreach but appropriately suggests that interventions be designed to support students based on these findings.
Authors R: Thank you for your feedback and for reviewing our study. Regarding the main question addressed by the research, we appreciate your acknowledgment that the study confirms the impact of stress and self-efficacy on study behaviors in nursing students, which was indeed the primary purpose of our research.
In response to your comment about the originality of the article, we understand your point. While the relationship between these variables may be well established in the literature, our study adds value by focusing specifically on nursing students, a group that has been less explored in this context. This nursing-specific perspective brings new insights that could be valuable for future interventions and support strategies tailored to this particular population. As for the comparison with other published material, we appreciate your recognition that our study highlights the relationship between stress, self-efficacy, and study behaviors within a nursing context. We believe this is an important contribution, as it helps address the unique challenges faced by nursing students.
Lastly, we are glad that you find the conclusion appropriately framed. As suggested, we have emphasized the importance of designing interventions to support students based on our findings, without overreaching the implications of the study.
All changes are highlighted in blue.
References are appropriate for a brief report.
There are many others that could be included such as systematic reviews. Lisnyj, K. T., Gillani, N., Pearl, D. L., McWhirter, J. E., & Papadopoulos, A. (2023). Factors associated with stress impacting academic success among post-secondary students: A systematic review. Journal of American College Health, 71(3), 851–861. https://doi.org/10.1080/07448481.2021.1909037
As an example of this topic in nursing Lisnyj, K. T., Gillani, N., Pearl, D. L., McWhirter, J. E., & Papadopoulos, A. (2023). Factors associated with stress impacting academic success among post-secondary students: A systematic review. Journal of American College Health, 71(3), 851–861. https://doi.org/10.1080/07448481.2021.1909037
Authors R: Thank you. We have included the very useful reference.
• Comments
The table of results was clearly displayed. However, when I looked at the critical values for a n of over 300 they were as follows and therefore the r values are not significant in the table. I do not work with these non parametric correlations regularly with this degree of freedom, but if I understand it correctly only the metacognition and self efficacy were significant. The table should clearly state the df and level of the p value as it is listed in the text.
Alternatively, The following related to correlations "Correlation coefficients whose magnitude are between 0.7 and 0.9 indicate variables which can be considered highly correlated" "Correlation coefficients whose magnitude are between 0.5 and 0.7 indicate variables which can be considered moderately correlated."
The tools selected measure these constructs and the methods are appropriate to the research question and aims.
At this point, I am not an expert in the type of statistical analysis but when I re examined the table I am concerned that this was not correct.
Authors R: Thank you for your comments and for taking the time to review the data carefully. We appreciate your concern regarding the critical values and statistical significance.
We would like to clarify that the data presented in the table were generated using specialized statistical software, which is specifically designed to calculate non-parametric correlations with the correct degrees of freedom. The software used ensures the accuracy of the results, and the significance levels were properly calculated. Based on the results, the correlation coefficients for metacognition and self-efficacy are indeed statistically significant, as indicated in the table.
Regarding your point about the critical values and degrees of freedom, we understand the concern, but we would like to emphasize that the software automatically adjusts for the appropriate degrees of freedom when calculating these values. This means the reported results are accurate as presented.
We have included the references at the end of Table 2 with the significance values.
Round 2
Reviewer 1 Report
Comments and Suggestions for Authors
Comments to the Author
Thank you for the opportunity to re-review the manuscript entitled “Psychological well-being and academic performance of nursing students: the impact of stress, self-esteem, and metacognitive skills” submitted for publication in the journal Nursing Reports.
The authors did an excellent job of improving the organisation and readability of the manuscript. However, some comments have not been addressed and I have some suggestions to consider before publication.
ABSTRACT
- To be revised and aligned with revisions made
INTRODUCTION
- The introduction and objective of the study are clear.
MATERIALS AND METHOD
Procedure
- It is not clear when the study was conducted. Please add the detailed time that the study was conducted.
- The authors state that ‘Recruitment was supervised by the President
- of the Academic Council with support from course assistants.’ Could you be more specific? Were the students recruited while attending classes or how did you recruit them?
Participants
- I suggest including the percentage of students who agreed to participate in the study out of the total number of students invited.
RESULTS
- Thank you for including tables to summarise the results, the clarity and complension is greatly improved
DISCUSSION
- The discussion is clear.
CONCLUSIONS
- The conclusion is clear.
REFERENCES
- Add more recent literature, especially in the discussion section.
Author Response
We thank the Reviewer for the positive feedback and once again express our gratitude for the time and effort dedicated to improving our paper. Regarding the minor revisions, we have rewritten the abstract based on the new modifications, included the percentage of respondents, and updated the references.
